# Multiscale Simulation on the Thermal Response of Woven Composites with Hollow Reinforcements

**DOI:** 10.3390/nano12081276

**Published:** 2022-04-08

**Authors:** Xiaoyu Zhao, Fei Guo, Beibei Li, Guannan Wang, Jinrui Ye

**Affiliations:** 1School of Mechanical and Automotive Engineering, Shanghai University of Engineering Science, Shanghai 201620, China; zhaoxiaoyu@sues.edu.cn (X.Z.); gf13383897368@outlook.com (F.G.); chancbllst@163.com (B.L.); 2Department of Civil Engineering, Zhejiang University, Hangzhou 310058, China; 3School of Aerospace Engineering, Beijing Institute of Technology, Beijing 100081, China

**Keywords:** hollow material, progressive homogenization, thermal conductive behavior, plain-woven composites

## Abstract

In this paper, we established a progressive multiscale model for a plain-woven composite with hollow microfibers and beads and investigated the general conductive thermal response. Micromechanic techniques were employed to predict the effective conductivity coefficients of the extracted representative volume elements (RVEs) at different scales, which were then transferred to higher scales for progressive homogenization. A structural RVE was finally established to study the influence of microscale parameters, such as phase volume fraction, the thickness of the fibers/beads, etc., on the effective and localized behavior of the composite system It was concluded that the volume fraction of the hollow glass beads (HGBs) and the thickness of the hollow fibers (HFs) had a significant effect on the effective thermal coefficients of the plain-woven composites. Furthermore, it was found that an increasing HGB volume fraction had a more significant effect in reducing the thermal conductivity of composite. The present simulations provide guidance to future experimental testing.

## 1. Introduction

Hollow materials and structures have been widely used in the aerospace, marine and energy fields due to their desirable properties, including high strength-to-weight ratios, large contact areas, etc. In addition, it has been shown that those hollow microstructures can be employed as either natural or human-made heat-insulating materials. For instance, it is common to embed hollow fibers/beads in more sophisticated material systems to improve their thermal insulating performance by taking advantage of the air’s extremely low thermal conductivity. To avoid laborious and costly experimental measurements, distinct micromechanics techniques have been developed to not only predict, but more importantly design, structural materials. Hollow structural fillers have some inherent advantages, such as high strength-to-weight ratios and better heat and noise insulation, endorsing their extensive applications in the aerospace industry, civil structures, vehicle innovations, etc. [1,2,3]. As early as 1994, May et al. [4] manufactured hollow diamond fibers that were demonstrated to possess almost the same strength as traditional solid fibers. They also suggested that the new fibers could be more functional if they were filled with proper materials. Raudenský et al. [5] prepared and tested two liquid-to-air cross-flow polymeric hollow fiber heat exchangers to rival the traditional finned-tube heat exchangers and found that the total heat transfer coefficient of the hollow fiber cross-flow heat exchanger could reach 200~450 W/(mK). With the aggravation of global warming and the energy shortage, innovative heat insulating materials could provide an attractive alternative to effectively alleviate those issues. It has been demonstrated that refined micro-materials, such as hollow fibers/beads, are intriguing options [6,7].

Relative to traditional composites, the mechanism of heat transmission becomes more complex in a composite system with multiple phases and voids; this has drawn attention from researchers who have conducted relevant investigations on their thermal behavior [8,9,10]. Ren et al. [11] prepared composites using borosilicate glass (BG) and hollow glass beads (HGBs) as matrix and filler, respectively, and found that the effect of the solid phase appeared to not be important for materials with low thermal conductivity and with a porosity greater than 70%. Zhu et al. [12] analyzed the thermal conductivity, dielectric constant, loss and compressive moduli and strength of composites filled with four types of HGBs. They also proposed that the composite’s properties could be controlled by tailoring the volume fraction and density of the HGBs, which offered a more extensive prospect of HGBs in composite systems. Luo et al. [13] prepared flexible paraffin/MWCNTs/PP hollow fiber membrane multi-phase composite materials, whose shape stability, heat storage performance and heat-conducting properties were analyzed. Liu et al. [14] studied the effective thermal behavior of HGB-reinforced composites using a transient plane source method and a numerical method. Xing et al. [15] designed a lightweight and thermally insulated composite material with HGBs as filler. It was demonstrated that the thermal behavior and bending response could be improved by tailoring the micro-parameters of the HGBs. 

In addition to regular composite systems, woven composites have become attractive options as they can sustain not only in-plane tension but also out-of-plane loads. Research has shown that embedding hollow fibers in the woven systems can improve their thermal insulating performance [16]. With the advancement of material science, researchers have realized that employing hollow fillers can also improve the thermal behavior of woven braided materials. The inhomogeneities embedded in woven composites can be manufactured or natural, such as plant fibers or animal hair [16,17,18,19,20,21]. For instance, Lin et al. [17] manufactured thermally insulated woven composites using PET non-woven fabrics and bamboo charcoal woven fabrics. Inspired by animals in extremely cold environments, Cui et al. [18] used a “freeze-spinning” technique to produce a bionic fiber braided composite material with polar bear hair. Kamble et al. [21] designed hollow braided composites for aircraft wings and wind turbines and prepared unidirectional 2D and 3D braided composites to evaluate their mechanical properties under static and dynamic loading.

In previous work in the literature, most of the research objects were focused on thermal composites filled with HFs or HGBs; their thermal conducting performance was predicted through experimental measurement. However, studies focusing on the quantitative and numerical analysis of woven fabric with hollow reinforcement are scarce. Our novelties are as follows: Firstly, this paper proposed a progressive multiscale simulation framework for the thermal responses of plain-woven braided composite reinforced with HGBs and hollow fibers and then studied the effects of structural and material properties within microstructures on global and localized thermal behavior. Then, considering the efficiency of analysis, we introduced the locally exact homogenization theory (LEHT). By extracting representative volume elements (RVEs) at different scales, the homogenized conductivity coefficients of each RVE were generated using finite element (FE) or LEHT through solving the governing equation of heat conduction and applying periodic boundary conditions. Those effective coefficients were then treated as input data for higher level homogenization.

## 2. Methods

The thermal conductive behavior of hollow woven composites depends on the composition of their microstructure and the conductivities of their multi-phases. In this work, we developed a progressive multiscale homogenization model towards this end, whose fundamental scheme is illustrated in Figure 1.

Herein, we defined several RVEs at different scales for the progressive homogenization of hierarchical composite structures. At the RVE_1_ scale, a representative volume element of the HGBs was established, and we investigated its thermal conductivity with different volume fractions of microbeads (10%, 20%, 30% and 40%). At the RVE_2_ scale, the representative volume element of the yarn (RVE_2_) was established, and LEHT [22,23] was adopted to study the thermal conductivity of RVE_2_ with different thicknesses of fiber. Finally, based on the aforementioned homogenized data, a representative unit cell of composites was established, and the effective thermal conductivities of RVE_3_ with different HGB volume fractions and thicknesses of hollow fiber were studied using the finite element method (FEM).

### 2.1. RVE_1_ with Hollow Glass Beads (HGBs)

HGBs are usually prepared as innovative fillers within composite systems to effectively improve their mechanical and thermal behavior (Figure 2a). Usually, the required HGBs are washed with ethanol, dried in a drying box and then mixed within the matrix. In actual production, the effective behavior of composites can be varied by tailoring the parameters at microscales, such as the phase volume fraction and thickness of the glass beads, to fulfill certain requirements in the material’s applications. To test the geometrical and material parameters, Figure 2b establishes an RVE_1_ model with HGBs.

FE-based ABAQUS 2020, which is a product of Dassault Systemes Simulia Corp., Johnston, RI, USA, was adopted to simulate the thermal behavior of RVE_1_. Here, it was assumed that the HGBs were uniformly distributed in the matrix domain in a cubic fashion; thus, the periodic boundary conditions had to be imposed on the opposite surfaces of the cubic element. The extracted RVE_1_ consisted of three constituents: air, glass shell and pure matrix. The volume fraction $v_{B}$ and the thickness of the HGBs $t_{B}$ were treated as the designed parameters, where the former is defined as $v_{B}=4\pi r_{1}{}^{3}/3l_{1}{}^{3}$, in which $l_{1}$ is the length of RVE_1_ and $r_{1}$ is the radius of an HGB.

### 2.2. RVE_2_ with Hollow Fibers

This section establishes an RVE_2_ model. Distinct from the traditional woven materials with solid fibers, the present work considered yarn reinforced with hollow fibers, as shown in Figure 3a. Taking advantage of the sake of heat insulation and weight reduction provided by voids, RVE_2_ was composed of three constituents in the simulation, including matrix, fiber and air, as shown in Figure 3b. It should be noted that the matrix in RVE_2_ is actually the “homogenized” matrix, whose thermal conductivity was obtained from the homogenization of RVE_1_ in the last section. Similar to the last sub-section, the hollow fibers were assumed to be uniformly distributed with the yarn in a hexagonal fashion, and the HF’s volume fraction was defined as follows:(1)$$v_{f}=\pi (r_{2}^{o}{}^{2}-r_{2}^{i}{}^{2})/l_{2}^{a}l_{2}^{b}$$
where $l_{2}^{a}$ and $l_{2}^{b}$ represent the length and width of RVE_2_, respectively, and the inner and outer radius of an averaged hollow fiber are represented by $r_{2}^{i}$ and $r_{2}^{o}$, respectively. The recently developed elasticity-based local exact homogenization theory (LEHT) was employed to study the thermal behavior of RVE_2_. Here, we will briefly rephrase the key steps of LEHT. The detailed steps can be seen in [22].

The main idea of LEHT is to adopt the Trefftz concept, which indicates that the thermal behavior is directly obtained as an explicit analytical solution through directly solving the partial governing equations of the heat conduction of a media as follows [23]:(2)$$T^{\lambda }=\sum_{n=0}^{\infty }a[(\xi^{n}H_{n1}{}^{\lambda }+\xi^{-n}H_{n3}{}^{\lambda })\cos n\theta +(\xi^{n}H_{n2}{}^{\lambda }+\xi^{-n}H_{n4}{}^{\lambda })\sin n\theta ]$$
(3)$$\begin{array}{c} q_{r}{}^{\lambda }=-k^{\lambda }({\bar{H}}_{2}\cos\theta +{\bar{H}}_{3}\sin\theta )-k^{\lambda }\sum_{n=1}^{\infty }n[(\xi^{n-1}H_{n1}{}^{\lambda }-\xi^{-n-1}H_{n3}{}^{\lambda })\cos n\theta \\ +(\xi^{n-1}H_{n2}{}^{\lambda }-\xi^{-n-1}H_{n4}{}^{\lambda })\sin n\theta ] \end{array}$$
where λ=hf,m represent the hollow fibers and matrix, respectively, and ξ=r/a is a dimensionless parameter. $H_{nt}{}^{\lambda }(t=1,\ldots ,4)$ are the unknown coefficients in the expression that can be determined by first imposing the interfacial boundary conditions of the temperature field and radial heat flux component between adjacent constituents:(4)$${\left.T^{hf}\right|}_{r=b}={\left.T^{m}\right|}_{r=b}\;,\;{\left.q_{r}^{hf}\right|}_{r=b}={\left.q_{r}^{m}\right|}_{r=b}$$
(5)$${\left.q_{r}^{hf}\right|}_{r=a}=0$$

The rest of the unknown coefficients can be determined through periodic boundary conditions [23]:(6)$$\pi =\frac{1}{2}\int_{V}q_{i}H_{i}dV-\int_{{}_{S_{Q}}}Q^{0}TdS-\int_{S_{T}}T^{0}QdS$$
where $Q^{0}$ and $T^{0}$ are the periodic heat flux and temperature, respectively, and Q is the normal component of the surface heat flow $q_{n}=q_{i}n_{i}$ along the surface $S_{k}$ as follows:(7)$$Q(S_{k})=\int_{S_{k}}q_{i}n_{i}dS$$
where $n_{i}$ is the unit vector normal to the surface $S_{k}$.

### 2.3. RVE_3_ Simulation

Finally, an RVE_3_ model was established for the plain-woven composite (see Figure 1 for details), whose basic composition included a homogenized matrix and four mutually orthogonal fiber bundles, which were assumed to be uniformly distributed. Those bundles also possessed homogenized properties from the simulation of RVE_2_. The mesh discretization of the geometric model was conducted in TexGen, and then transferred to ABAQUS for numerical simulation. TexGen (v3.12.0, University of Nottingham, Nottingham, UK) is an open-source software licensed under the General Public License that was developed at the University of Nottingham for modelling the geometry of textile structures and has been used by the Nottingham team as the basis of models for a variety of properties, including textile mechanics, permeability and composite mechanical behavior. 

### 2.4. Homogenization at Distinct Levels

FE and elasticity-based numerical or semi-analytical models were employed to solve the partial governing equations for heat conduction, after which the homogenized Fourier law was established to obtain the effective coefficients. We will describe a few key steps.

Here, we start from the well-known Fourier’s law: $Q_{i}=-k_{ij}A\Delta T/\Delta x_{j}$, where $Q_{i}(i=1,2,3)$ is the homogenized heat flux component of the unit cell, $k_{ij}$ is the effective thermal conductivity of the unit cell in different directions, A is the area of the unit cell through which heat flux passes and $\Delta x_{i}$ is the characteristic length of the RVE in the three directions that we impose the heat gradients ΔT. From Section 2.1, Section 2.2 and Section 2.3, the local distributions of heat flux density $q_{i}$ can be easily obtained after imposing a heat gradient at opposite surfaces of each RVE, whose thermal/heat flux distributions are generated through the FE or elasticity-based simulations introduced in Section 2.1, Section 2.2 and Section 2.3, from which the homogenized heat flux components are calculated as [23] follows:(8)$$\bar{q}=\frac{1}{V}\sum_{\lambda }\int q^{\lambda }(x)dV_{\lambda }=\sum_{\lambda }v_{\lambda }{\bar{q}}^{\lambda }$$
where $v_{\lambda }$ represents the volume fraction of the fiber/air/matrix in the yarn, V represents the volume of the entire unit cell and $q^{\lambda }(x)$ represents the macroscopic heat flux densities within any phase λ=p, f, m of RVE*_i_* (*i* = 1,2,3).

Therefore, Fourier’s law can be explained in a more detailed form to calculate the effect of the thermal conductivity coefficients in three directions [23] as follows:(9)$$\bar{q}=k^{\mathrm{eff}}\bar{H}$$
where $k^{\mathrm{eff}}$ is the effective thermal conductivity of RVE*_i_*, $\bar{H}$ is the average temperature gradient of the material and ${\bar{H}}_{j}=\Delta T/\Delta x_{j}$. The conductivity coefficients are finally obtained for each RVE*_i_*, and the woven composites of a lower-level RVE are then treated as input construction for higher-level RVEs, giving the final effective conductivities of the composites with hollow reinforcements. We will validate our model in the next section.

## 3. Results

We employed the RVEs across several levels and established numerical models to investigate the local thermal distributions and effective conductivity coefficients of woven composites. The simulated results were validated against existing micromechanics in the literature.

Here, we started from the RVE_1_ featuring a matrix embedded with HGBs. We tested the effect of volume fraction on the thermal conductivity by setting $v_{B}$ equal to 10%, 20%, 30% and 40%. The averaged inner and exterior diameters of an HGB were 57.04 μm and 58.64 μm, respectively. The thermal coefficients of the constituents are listed in Table 1. Tetrahedral mesh elements (DC3D10) were employed in the simulation. Taking $v_{B}=20\%$ as an example, 126,033 nodes and 78,399 elements were required for the simulation. By imposing a heat gradient and boundary conditions on RVE_1_, the local thermal distributions were determined, as shown in Figure 4.

In order to verify the accuracy of the established simulation model, the present results were compared with the existing analytical classical micromechanics models, including the thermal-electric analogy [24], Hashin-Shtrikman (H-S) model and effective media seepage theory (EMPT), which considers the random mixture of two homogeneous phases via a continuous pore path [25,26]. Table 2 shows the comparison of our results with the results from Liu et al. [14], who also employed an FE-based simulation and the classical models mentioned above. For the reader’s interest, those equations are listed below (10)–(12):(10)$$\lambda_{\mathrm{eff}1}^{P}=v_{B}\lambda_{\mathrm{HGB}}+(1-v_{B})\lambda_{m}\;(\mathrm{Parallel})$$
(11)$$\lambda_{{}_{\mathrm{eff}1}}^{\mathrm{H-S}}=\lambda_{m}\left[\frac{\lambda_{\mathrm{HGB}}+2\lambda_{m}+2v_{B}(\lambda_{\mathrm{HGB}}-\lambda_{m})}{\lambda_{\mathrm{HGB}}+2\lambda_{m}-v_{B}(\lambda_{\mathrm{HGB}}-\lambda_{m})}\right]\;(\mathrm{H-S})$$
(12)$$\lambda_{{}_{\mathrm{eff}1}}^{\mathrm{EMPT}}=\frac{1}{4}\left[\lambda_{\mathrm{HGB}}(3v_{B}-1)+\lambda_{m}(3v_{m}-1)+\sqrt{{\left[\lambda_{\mathrm{HGB}}(3v_{B}-1)+\lambda_{m}(3v_{m}-1)\right]}^{2}+8\lambda_{m}\lambda_{\mathrm{HGB}}}\right]\;(\mathrm{EMPT})$$

By comparison with results in the literature, it can be observed that all the models generate well-matched results, especially when considering the results reported by Liu et al. [14] and the thermal-electric analogy. A few discrepancies were obtained when considering the analytical micromechanics model, likely due to the fact that the authors employed a few simplified assumptions.

The effective coefficients generated from the RVE_1_ simulation were further treated as input data to the RVE_2_ to represent the “homogenized” matrix. In this model, the thermal conductivity of the fiber was $11\;W/\left(\mathrm{mK}\right)$. Taking the volume fraction of HGBs as $v_{B}=10\%$, LEHT was used to study the thermal response of yarn with different fiber thicknesses when the hollowness ratio (the proportion of the volume of air in RVE_2_) was 30%.

An FE simulation was employed to verify the thermal conductivity of RVE_2_ as predicted by LEHT. The thickness of the RVE_2_ simulation was 1 μm. The thickness of hollow fiber was 1.5 μm, and the inner radius was 3 μm. We still adopted DC3D10 in the FE simulation, where a total of 21,937 nodes and 12,211 elements were required. As was indicated in Section 2.4, the thermal conductivity coefficients of RVE_2_ can be predicted in accordance with the effective Fourier law by imposing the temperature gradients in different directions. Table 3 compares the results calculated by LEHT and FE; it can be seen that good agreement was still obtained. In addition, it can be easily seen from Figure 5b that the heat flow was concentrated at the vicinity of the fiber domain; the dark blue part in the picture represents a very low heat flow due to the minimum thermal conductivity of air in the hollow fiber.

Finally, the homogenized coefficients of RVE_2_ were further transferred to RVE_3_ to predict the effective response of plain-woven composites. The volume fraction of the fiber yarn in the model was $v_{y}=55.2\%$. The parameters of the established geometric model are listed in Table 4. The FE analysis of RVE_3_ was illustrated by taking $v_{B}=10\%$ in the matrix and the hollowness ratio in the RVE_2_ as 30% (as shown in Figure 6). A total of 174,978 elements and 275,010 nodes were used.

The FEM results from this study were also compared with the analytical model offered by Lewis and Nielsen [27], which is known as the L-N model. Zhou et al. [28] offered an explicit expression of the L-N model for the effective thermal conductivity of braided composite materials as $\lambda_{\mathrm{eff}}^{\mathrm{L-N}}$ ($\phi_{m}$ = 1, ξ = 0.434):(13)$$\lambda_{\mathrm{L-N}}=k_{m}\frac{1+\xi \beta v_{y}}{1-\psi \beta v_{y}},\;\beta =\frac{k_{y}/k_{m}-1}{k_{y}/k_{m}-1+\xi },\;\psi =1+\frac{1-\varphi_{m}}{\varphi_{m}^{2}}v_{y}\;\lambda_{\mathrm{eff}}^{\mathrm{L-N}}=k\frac{1+0.434\beta v_{y}}{1-\beta v_{y}},\;\beta =\frac{k_{y}-k_{m}}{k_{y}+0.434k}$$
where ξ and $\varphi_{m}$ are location parameters related to the fiber bundle shape and $k_{y}$ and $k_{m}$ are the thermal conductivity of the yarn and matrix, respectively.

Table 5 compares the present results and the L-N expression with almost no discrepancy, demonstrating the good credence of the present model. What may even be considered superior is that the present simulation can recover the local thermal distributions within RVE_3_ where the heat flow is concentrated in the contacting part between the matrix and the yarn.

## 4. Discussion

Due to the complex structural configuration of braided composites, it is expensive to conduct direct experimental measurements to investigate the effect of geometrical and material parameters on the thermal behavior of braided composites with different compositions. This paper studied several micro-parameters and their effect on braided composites with hollow fillers based on a multi-scale simulation for the design of thermally insulating devices. Based on the calculation from the third part of this paper, we calculated the thermal conductivity of the hollow fibers with a corresponding fiber thickness; it was found that as the volume fraction of the HGBs increased from 10% to 40%, the thermal conductivity of the “homogenized” matrix in RVE_2_ was gradually reduced by about one-third of the original matrix. We also tested the effect of hollow fibers by employing several thicknesses, including 0.5 μm, 1.0 μm, 1.5 μm and 2.0 μm, whose corresponding volume fractions of the fiber were $v_{f}=40.10\%,\;53.3\%,\;67.5\%$ and 83.3%, respectively. From the analytical results of RVE_1_ and RVE_2_, the thermal insulating performance of the woven composite material with a matrix material filled with hollow glass balls and hollow fibers was improved. For this reason, we analyzed the thermal conductivity of solid fiber woven composites when the HGB volume fraction was 10%. The results of the macro-scale thermal conductivity of the fiber woven composite material are shown in Table 6; the results verified our conjecture very well. Therefore, we concluded that hollow structural fibers can effectively improve heat insulating performance.

Because the volume fraction of hollow glass beads plays a significant role in the varying thermal conductivity of the matrix material, and in the woven structure composite material, both the RVE_1_-scale and RVE_2_-scale contained the matrix material; thus, the volume fraction of the hollow glass beads may also affect the macroscopic thermal conductivity of the material. In order to further study its specific effects, we used the LEHT model to calculate the thermal conductivity of yarn with different HGB volume fractions; these values were used as data for the homogenized matrix of yarn within different hollow fiber shell thicknesses as analyzed using the FEM. In order to visually express the change of the effective thermal conductivity with the volume fraction of the HGBs and the shell thickness of the hollow fiber, the FEM analysis results are shown in Figure 7. It can be easily concluded that as the volume fraction of HGB increased, the RVE_3_-scale thermal conductivity with different fiber shell thicknesses was significantly reduced. This is because there is air included in the HGBs. As $v_{B}$ increases, the air content also increases. In addition, as the shell thickness of the hollow fiber decreased, the fiber volume fraction decreased, and the coefficient was also significantly reduced. This is because the increase in the shell thickness of the hollow fiber means that the volume fraction of air in the composite material increases, and its extremely small thermal conductivity causes the overall thermal conductivity of the composite material to decrease. In summary, the thermal insulation performance of the material was positively correlated with the volume fraction of the hollow glass spheres, and negatively correlated with the shell thickness of the hollow fibers. 

Besides the effective behavior, the localized response of the woven composites within RVEs at different scales was recovered, which was critical for identifying possible crack and propagation. Figure 8 shows the recovery of the local temperature and heat flux distributions within RVE_3_ under the temperature gradient ${\bar{H}}_{2}=1\;{}^{\circ}C$. The averaged HF thickness was 1.5 μm with a hollowness ratio of 0.3. It can be seen from the figure that the heat flux concentration was generally located at the junction of the matrix and the fiber due to the mismatch of thermal coefficients. We also observed that the value of heat flux density decreased with an increasing volume fraction of HGBs. When the volume fraction of the hollow glass spheres was increased from 10% to 40%, the maximum heat flow of the yarn decreased during the heat conduction process of the composite material. The heat flow density was reduced from 6.487 $W/m^{2}$ to 3.754 $W/m^{2}$. In addition, it can be seen in the temperature distribution of Figure 9 that, along the z-axis direction, the isothermal surface was parallel to the coordinate plane, and along the y-axis direction, the isothermal surface was not completely parallel to the coordinate plane, instead fluctuating slightly along the fiber bundle direction.

## 5. Conclusions

In this work, we proposed a multiscale simulation model to study the thermal behavior of woven composites with hollow reinforcements. Several microscale parameters were varied to numerically test their influence on the effective and localized responses of the composite system. The main conclusions obtained in this paper are as follows: (1) The volume fraction of the hollow glass microsphere filler had a significant effect on the macroscopic thermal conductivity of the composite material. As the volume fraction of the hollow glass microsphere increased, the macroscopic thermal conductivity of the composite material decreased. (2) The thickness of the hollow fibers also impacted the tailoring of the effective thermal conductivity of composite materials by changing the fiber volume fraction of RVE_2_. (3) The efficient LEHT theory was employed to effectively predict the thermal conductivity of yarn, and was demonstrated as an efficient tool to replace the FEM simulation that will also be introduced in more general micromechanics backgrounds.

The analysis in this paper was mainly based on the existing theory to simulate and predict the macroscopic thermal conductivity of plain-woven composites. In future work, the actual measurement of the sample will be carried out.

## Figures and Tables

**Figure 1 nanomaterials-12-01276-f001:**
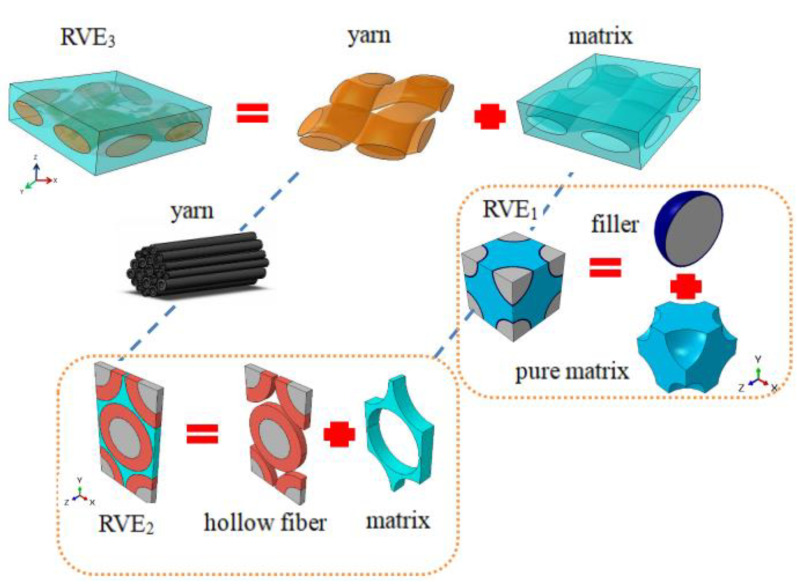
Progressive modeling framework of woven composites with HGBs and HFs.

**Figure 2 nanomaterials-12-01276-f002:**
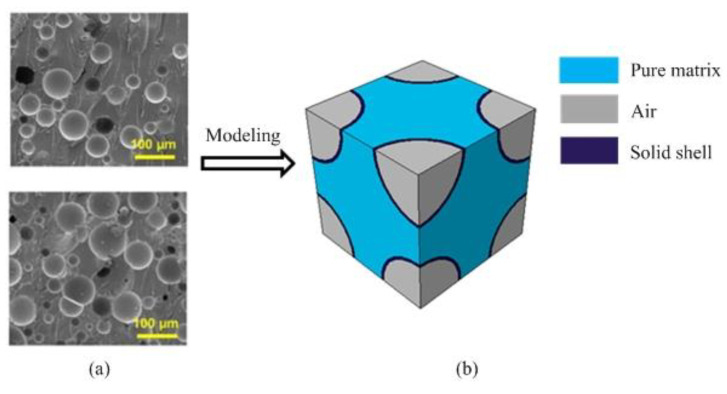
(**a**) SEM images of composites with different HGB contents (reprinted with permission from Ref. [15]. Copyright 2020 Elsevier) and (**b**) the corresponding theoretical RVE_1_ model with matrix, shell and air.

**Figure 3 nanomaterials-12-01276-f003:**
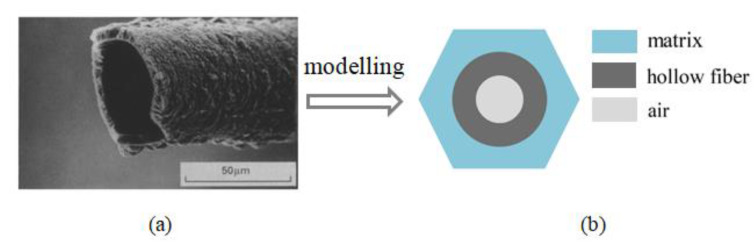
(**a**) SEM images of a hollow fiber (reprinted with permission from Ref. [4]. Copyright 1994 Elsevier) and (**b**) the corresponding RVE_2_ model with matrix, fiber phase and air.

**Figure 4 nanomaterials-12-01276-f004:**
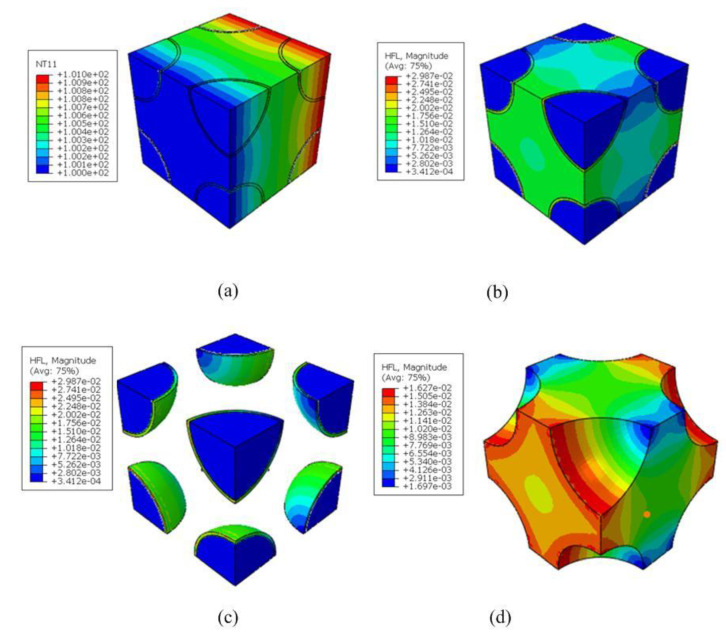
Distributions of the (**a**) temperature and (**b**) heat flow of RVE_1_ along the y−direction; (**c**) heat flow of HGBs along the y−direction; and (**d**) heat flow of pure matrix with $v_{B}=20\%$
along the z−direction.

**Figure 5 nanomaterials-12-01276-f005:**
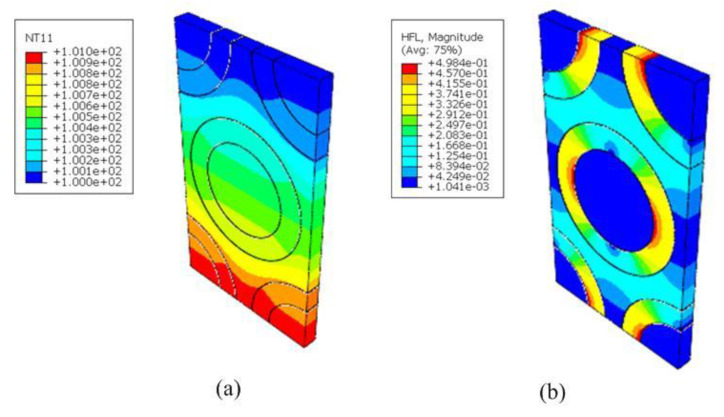
Distribution of the (**a**) temperature and (**b**) heat flow along the y−direction.

**Figure 6 nanomaterials-12-01276-f006:**
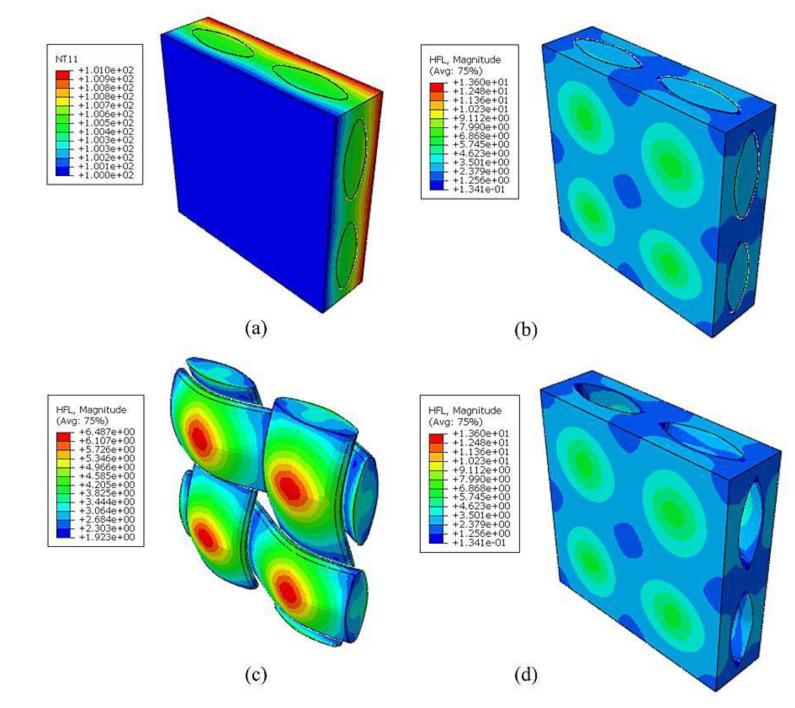
Distributions of (**a**) temperature, (**b**) the heat flow of RVE3, (**c**) the heat flow of the yarns and (**d**) the heat flow of the matrix under a temperature gradient
$${\bar{H}}_{j}=1\;{}^{\circ}C$$
in the z−direction.

**Figure 7 nanomaterials-12-01276-f007:**
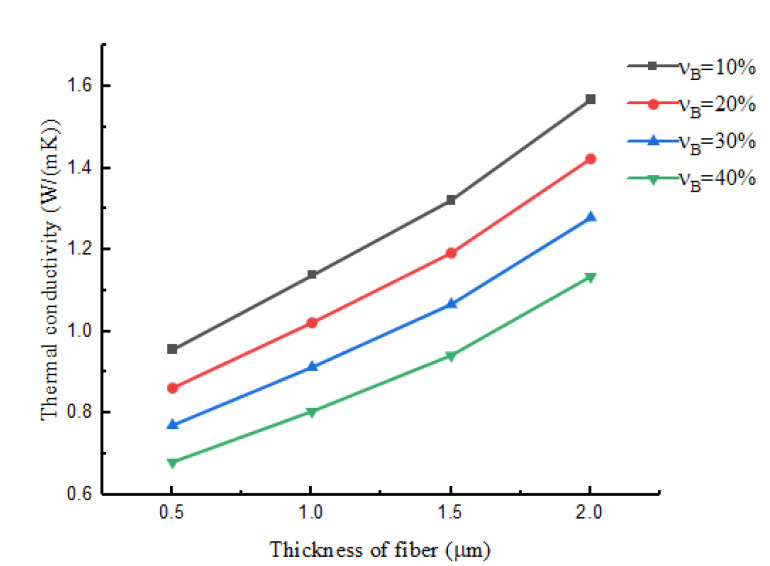
Predicted effective thermal conductivities with different volume fractions of HGBs and thicknesses of hollow fiber.

**Figure 8 nanomaterials-12-01276-f008:**
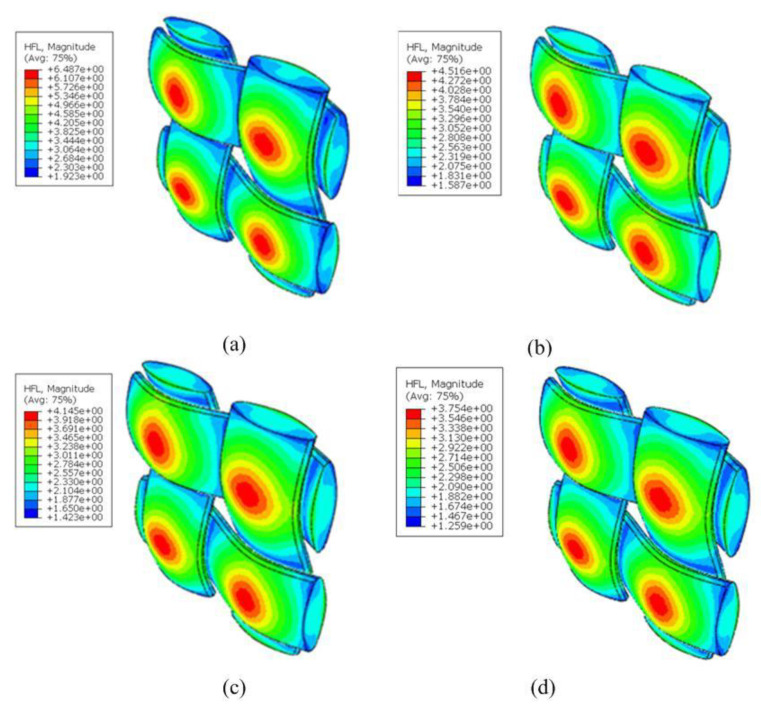
Distribution of heat flow of yarn with different volume fraction of HGBs: (**a**) $\nu_{B}=10\%$, (**b**)
$\nu_{B}=20\%$, (**c**)
$\nu_{B}=30\%$, (**d**)
$\nu_{B}=40\%$.

**Figure 9 nanomaterials-12-01276-f009:**
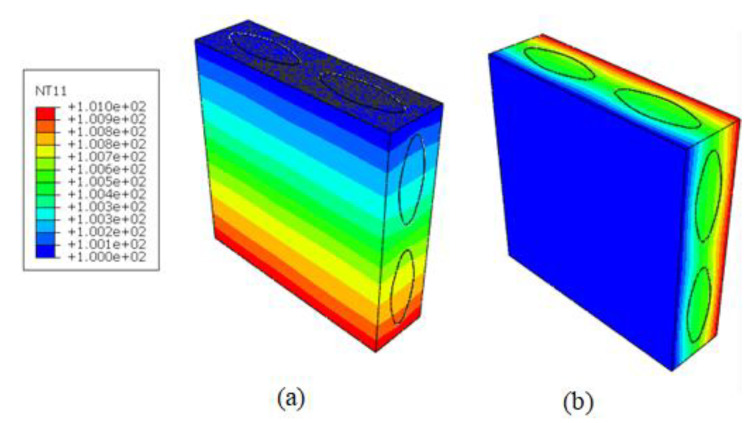
Distribution of temperature along the (**a**) y−axis direction and (**b**) z−axis direction.

**Table 1 nanomaterials-12-01276-t001:** Thermal conductivity coefficients of compositional phases within RVE_1_.

Phase	Air	Solid	Matrix
Value/(W/(mK))	0.023	1.03	0.93

**Table 2 nanomaterials-12-01276-t002:** Comparison of present conductivity coefficients of RVE_1_ with results in the literature.

Conductivity Coefficients [Unit: W/(mK)]	$v_{B}$			
	10%	20%	30%	40%
$\lambda_{\mathrm{eff}1}^{\mathrm{Present}}$	0.8240	0.7274	0.6358	0.5484
$\lambda_{{}_{\mathrm{eff}1}}^{\mathrm{H-S}}$ [11]	0.8018	0.68481	0.5777	0.4793
$\lambda_{{}_{\mathrm{eff}1}}^{\mathrm{EMPT}}$ [11]	0.7959	0.6625	0.5305	0.4010
$\lambda_{\mathrm{eff}1}^{\mathrm{Liu}}$ [14]	0.8240	0.7254	0.6333	0.5468
$\lambda_{\mathrm{eff}1}^{P}$ [24]	0.8393	0.7486	0.6579	0.5672

**Table 3 nanomaterials-12-01276-t003:** Thermal conductivity of yarn with different fiber thicknesses ($r_{2}^{o}$ − $r_{2}^{i}$ ).

Thickness of Fiber/μm	0.5	1.0	1.5	2.0
$\lambda_{\mathrm{eff}2}^{\mathrm{LEHT}}$/[W/(mK)]	1.0973	1.5633	2.2148	3.4659
$\lambda_{\mathrm{eff}2}^{\mathrm{FEM}}$/[W/(mK)]	1.0986	1.5783	2.2219	3.4549
Errors	0.12%	0.95%	0.32%	0.32%

**Table 4 nanomaterials-12-01276-t004:** Geometric model parameters of plain-woven composites.

Parameter	Length	Thickness (*h*)	Height of Yarn (*c*)	Width of Yarn (*w*)	Distance between Adjacent Yarn
Value/mm	1.6	0.44	0.2	0.6	0.8

**Table 5 nanomaterials-12-01276-t005:** Comparison of results between FEM and L-N models.

** Thickness/μm **	**0.5**	**1.0**	**1.5**	**2.0**
Present/(W/(mK))	0.9550	1.1371	1.3211	1.5672
L-N/(W/(mK)) [28]	0.9607	1.1427	1.3267	1.5526
Error	0.593%	0.497%	0.419%	0.945%

**Table 6 nanomaterials-12-01276-t006:** Thermal conductivity of RVE_3_.

$v_{f}$	$\lambda_{\mathrm{eff}3}^{s}$ $/\left(W/(\mathrm{mK})\right)$	$\lambda_{\mathrm{eff}3}^{h}$ $/\left(W/(\mathrm{mK})\right)$	Reduction
40.1%	1.1740	0.9550	18.65%
53.3%	1.3216	1.1371	13.96%
67.5%	1.5123	1.3211	12.64%
83.3%	1.7973	1.5672	12.80%

$\lambda_{\mathrm{eff}3}^{s}$ is the thermal conductivity of RVE_3_ with solid fibers and $\lambda_{\mathrm{eff}3}^{h}$ is the thermal conductivity of RVE_3_ with hollow fibers.

## Data Availability

All data, models, or code that support the findings of this study are available from the corresponding author upon reasonable request.

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
