# Peer review of "Multiscale Simulation on the Thermal Response of Woven Composites with Hollow Reinforcements"

_nanomaterials, 2022, doi:10.3390/nano12081276_

Round 1
Reviewer 1 Report
Dear Authors,
Your work seams interesting but some revisions are required as follows.
- Regarding novelty: please clearly point out the novelty of your work comparatively to already published papers, in the Introduction chapter.
- The Introduction chpater is too long and some of the paragraphs can be deleted, for example last paragraph which is not apprpriate for a manuscript sent to this Journal, and a lot of initial paragraphs which are not directly related to this paper.
- The manuscript must be rewritten according to Journal recommendations for authors, which include Title, Abstract, Introduction, Methods, Results, Discussion and Conclusions.
- Regarding Keywords: I suggest to replace or improve the Keyword "homogenization" and to insert Keywords about simulation models.
- Please explain why do you name these composites woven composites since in figure 1, for example it doesn’t appear to be any woven composite. If they are not woven what is the novelty of your work?
- Do not insert any equations which are not original without References. See equation (7), for example. Check the rest of them.
- Please clarify in Table 2 which results are original and for the rest insert proper References in Table. The same for Table 5 (References).
- Table 6: the codes for thermal conductivity are identical, you must change one code to diferentiate them.
- I suggest you to move the Validations Subchapter before Conclusions so you can compare at the end of the paper your all results to experimental or other authors’ ones.
- Please improve also the Conclusions chapter according to the above observations.
Author Response
We thank the reviewers for the careful reading of our manuscript, and the constructive comments aimed at improving the clarity and impact of the presentation. We have thoroughly addressed these comments in the revision, incorporating the majority of recommended changes as described below in our detailed point-by-point responses. We hope that our revisions are satisfactory.
Reviewer #1: Your work seams interesting but some revisions are required as follows.
1.Regarding novelty: please clearly point out the novelty of your work comparatively to already published papers, in the Introduction chapter.
Response: We have listed our novelty in the last paragraph of Introduction. In this paper, our novelties conclude:(1) We proposed progressive homogenization model of woven composites with hollow glass beads and hollow fibers to analyze its thermal behavior. (2) We investigated several microstructural factors on thermal conductivity of woven composites by finite element method and the local the locally exact homogenization theory. It should be pointed out that there is scarce investigation on the thermal woven composites with hollow microstructures from a micromechanics perspective.
2.The Introduction chapter is too long and some of the paragraphs can be deleted, for example last paragraph which is not appropriate for manuscript sent to this Journal, and a lot of initial paragraphs which are not directly related to this paper.
Response: We have deleted some content of Introduction (including the last paragraph) but still keep the correct logical sequence.
3. The manuscript must be rewritten according to Journal recommendations for authors, which include Title, Abstract, Introduction, Methods, Results, Discussion and Conclusions.
Response: We have restructured the article according to Journal recommendations for authors. We have replaced the titles “Progressive Modelling Framework”, “Validations” and “Numerical Investigations” with “Methods”, “Results” and “Discussion”, respectively.
4. Regarding Keywords: I suggest to replace or improve the Keyword "homogenization" and to insert Keywords about simulation models.
Response: Our article decided to adopt “progressive homogenization” instead the original “homogenization”. The progressive homogenization method is to establish the model of woven composites. Firstly, matrix in RVE1 is homogenization matrix with pure matrix and hollow glass beads. Then, fiber in RVE2 is "homogenization yarn" with hollow fiber and "homogenization matrix". "Progressive homogenization" is the core of our modeling simulation, so we kept this keyword.
5. Please explain why do you name these composites woven composites since in figure 1, for example it doesn’t appear to be any woven composite. If they are not woven what is the novelty of your work? Response: Woven composites have periodic structure and we selected RVE3 as a represent volume element. In figure 1, we give a progressive model of RVE3 to investigate its thermal behavior, so RVE3 is the representative of woven composites and our novelty.
6. Do not insert any equations which are not original without References. See equation (7), for example. Check the rest of them.
Response: We have inserted reference [23] in equation (8) [previous equation (7)] and checked the rest equations to ensure proper quotation.
7. Please clarify in Table 2 which results are original and for the rest insert proper References in Table. The same for Table 5 (References).
Response: In Table 2, we have inserted reference [11] for and , and reference [24] for. In Table 5, we inserted reference [28] for L-N model. We have also denote “present” for the original results.
8. Table 6: the codes for thermal conductivity are identical, you must change one code to differentiate them.
Response: We have made for RVE 3 with hollow fiber and for solid fiber in Table 6.
9. I suggest you to move the Validations Subchapter before Conclusions so you can compare at the end of the paper your all results to experimental or other authors’ ones.
Response: In this paper, we proposed the progressive homogenization model to investigate the thermal behavior of woven composites. To proof the validity our model, we need to firstly validate our model before employing it to conduct numerical investigation. So we still decide to place the Results before Discussions.
10. Please improve also the Conclusions chapter according to the above observations.
Response: According to the above observations, we have improved our Conclusions.

Reviewer 2 Report
This manuscript describes a multiscale simulation model developed to study thermal behavior of woven composites with hollow reinforcements. A progressive multiscale homogenization model was proposed. The set of different parameters were varied to evaluate their impact on physical properties of composites. It was found that the volume fraction of HGB has a significant effect on thermal conductivity of composite demonstrating the decrease in macroscopic thermal conductivity as the volume fraction of the hollow glass microspheres is increased.
The paper is nicely illustrated, very well formatted and written in good English. It represents highly valuable results for the readers. In addition, the authors have provided sufficient data from literature to make necessary comparisons.
The scope of research fits very well for publication in Special Issue “Nanomechanics and Plasticity” of Nanomaterials journal. The only minor changes need to be done (few typos, actually) before being accepted for publication:
1) Page 1, Line 44: “…refined micro-materials, such as hollow fibers/beads, are intriguing options” – looks like it should be “… have intriguing options”.
2) Page 7, Table 2, Line 237: “Comparison of predicted conductivity coefficients of RVE1again results in the literature” should be replaced by “Comparison of predicted conductivity coefficients of RVE1 against results in the literature”.
3) Table 2 contains the values with more than 5 significant digits, i.e. 0.577738. The values look too precise and could be probably rounded to several decimal places.
Reviewer 3 Report
Manuscript: Multiscale Simulation on the Thermal Response of Woven Composites with Hollow Reinforcements
The paper presents a multiscale simulations on thermal response of homogenized composites with hollow reinforcements. A validation of the model is first carried out, then some parametric studies are conducted to illustrate the influence of some parameters on thermal responses HGBs. The paper needs some corrections and it should be revised by addressing the following items:
- The abstract should be concise, brief and describe the main objectives of the paper and not as an introduction. For instance this paragraph can be removed to introduction or reformulated “Hollow materials and structures have been widely used in aerospace, marine as well as energy fields due to excellent properties such as high strength-to-weight ratios, large contact areas, In addition, it is realized that those hollow microstructures can be employed as heat-insulating materials, either as natural or human-made materials. For instance, it is popular to embed hollow fibers/beads in more sophisticated material systems to improve their thermal insulating performance by taking advantage of the air’s extremely low thermal conductivity. To avoid laborious and costly experimental measurement, distinct micromechanics techniques are developed to not only predict, but more importantly design structural materials”. So, the abstract should be improved and reformulated.
- In the passage from page 4 to page 5, an equation of the volume fraction of the fiver vf appears without numbering (please check the numbering).
- The Figure 7 is not indicated in the text (check it) “Predicted effective thermal conductivities with different volume fractions of HGBs and thicknesses of hollow fiber” Furthermore, explain why the thermal conductivity of HGBs is highest with the lowest vB?
- In table 6, the authors mention that is the thermal conductivity of RVE 3 with solid fiber and also is the thermal conductivity of RVE 3 with hollow fiber. Please use different notation to differentiate the one derived from solid fiber and with hollow fiber.
Round 2
Reviewer 1 Report
Dear Authors,
You made the possible improvements and responded to some observations. I propose your paper to be published in the revised form.